# Combination of Oncolytic Virotherapy with Different Antitumor Approaches against Glioblastoma

**DOI:** 10.3390/ijms25042042

**Published:** 2024-02-07

**Authors:** Alisa Ageenko, Natalia Vasileva, Vladimir Richter, Elena Kuligina

**Affiliations:** 1Institute of Chemical Biology and Fundamental Medicine, Siberian Branch of the Russian Academy of Sciences, Akad. Lavrentiev Ave. 8, 630090 Novosibirsk, Russia; nataly_vas@bk.ru (N.V.); richter@niboch.nsc.ru (V.R.); kuligina@niboch.nsc.ru (E.K.); 2LLC “Oncostar”, R&D Department, Ingenernaya Street 23, 630090 Novosibirsk, Russia

**Keywords:** glioblastoma, oncolytic viruses, chemotherapy, radiotherapy, immunotherapy, combination therapy

## Abstract

Glioblastoma is one of the most malignant and aggressive tumors of the central nervous system. Despite the standard therapy consisting of maximal surgical resection and chemo- and radiotherapy, the median survival of patients with this diagnosis is about 15 months. Oncolytic virus therapy is one of the promising areas for the treatment of malignant neoplasms. In this review, we have focused on emphasizing recent achievements in virotherapy, both as a monotherapy and in combination with other therapeutic schemes to improve survival rate and quality of life among patients with glioblastoma.

## 1. Introduction

Glioblastoma is a WHO grade IV glioma and is one of the most malignant and aggressive tumors of the central nervous system, accounting about 49% of all brain malignancies [1,2]. Despite active study of the mechanisms of development and drug resistance of brain tumors, patients with glioblastoma live an average 15 months from diagnosis [3]. The standard therapy for gliomas consists of the maximal safest possible surgical resection of the tumor, i.e., resection of as much tumor tissue as possible while maintaining normally functioning neurological conditions. Subsequent chemo- and radiotherapy is applied in accordance with tumor type, grade, and molecular characteristics [4]. Despite multimodal therapy, glioblastoma cannot be completely removed with surgery due to the invasive growth potential of this tumor type [5]. Moreover, glioblastoma cells form resistance to standard therapy; this is why tumor invariably recurs in approximately 80% of cases within 2–3 cm along the edge of the primary lesion [6,7].

Currently, modern treatment methods of curing glioblastoma are being created using the novel therapeutic approaches, including virotherapy, immunotherapy, check-point inhibitors, cancer vaccines, and adoptive cell therapy. Therapy by oncolytic viruses (OVs) shows encouraging results in clinical trials of glioblastoma treatment, permitting us to consider it as an opportunity to move closer to curing this type of malignant tumor. However, the use of OVs as single drugs sometimes lead to insufficient effectiveness of therapy for such treatment-resistant tumor types, including gliomas. This work reviews the most noteworthy studies of OVs application both in monotherapy and in combination with different therapeutic approaches against glioblastoma.

## 2. Virotherapy against Glioblastoma

Recent advances in understanding of cancer biology have stimulated the development of numerous therapeutic approaches, one of which is virotherapy. Therapy by oncolytic viruses is a promising method for malignant tumors of various histogenesis, both as a monotherapy and as a part of combination therapy [8].

OVs are designed based on different families and strains of natural viruses. OVs are selected according to various criteria; the main one is selectivity against tumor cells while avoiding normally functioning cells. Some viruses have an innate ability to target cancer cells, such as animal viruses (e.g., myxoma virus, parvovirus, Newcastle disease virus). Other viruses (e.g., HSV, poliovirus, adenovirus) have also the same ability to target tumor cells, but must be genetically engineered to reduce or eliminate virulence against human healthy cells, improving their safety and enhance antitumor activity [9].

Oncolytic viruses have wide-ranging effects that target tumors from different angles. First of all, they have a selective viral replication in tumor cells and dissemination of viral particles in tumors (Figure 1A). Many viruses multiply better in tumor cells, since the process of carcinogenesis leads to the same cellular changes as a viral infection (e.g., p53 inactivation, apoptosis inhibition) [10]. The second most important mechanism is direct cell lysis, induced by the natural life cycle of the virus. During infection and viral replication, lysis of tumor cells occurs with the release of viral particles that continue to cascade infect neighboring cells, thereby enhancing the therapeutic effect of the drug (Figure 1B) [11]. Another one of the main effects of OVs’ influence on cancer cells is immunovirotherapy. OVs induce a proinflammatory tumor microenvironment and subsequent antitumor responses to counteract the immune evasiveness of malignant cells [12]. In addition, OVs can turn an immunologically “cold” tumors into “hot” tumors; that is, they can make the tumor visible to the immune system (Figure 1C) [13]. Additionally, OVs can be armed with therapeutic immune modulators (e.g., GM-CSF, IL-12, IFN-α, PD-L1) for tumor gene therapy (Figure 1D). These modifications enable the targeting of the tumor microenvironment and make the tumor less resistant to therapy [14]. Also, it is known that OVs can infect tumor-associated stromal cells like endotheliocytes, leading to the destruction of the tumor vascularization system (Figure 1E) [15].

The described ways in which OVs act on tumor masses confirm their potential as antitumor agents. Some OVs are already at different stages of clinical trials or have already been approved for use in clinical glioblastoma therapy (Table 1).

### 2.1. Herpes Simplex Virus

Herpes simplex virus type 1 (HSV-1) is an enveloped, double-stranded DNA virus that possesses many advantages in use as an oncolytic virus [37]. HSV-1 has a large genome, which enables the insertion of various transgenes to enhance the antitumor activity and immunogenicity of OVs. Also, the virus has a wide tropism for multiple cell types and its DNA does not integrate into the host genome [38]. Given the attractiveness of this virus for researchers, various OVs have been developed on its basis, targeting different types of malignant neoplasms, including glioblastoma.

One of these viruses is G207, created in the basis of HSV-1 by deleting both copies of the *γ34.5* and inserting the *E. coli lacZ* into the region of *ICP6*. *γ34.5* is a neurovirulence gene. When its deleted, HSV-1 cannot complete lytic infection in neurons and thus does not cause encephalitis. Upon inactivation of *ICP6*, encoding a large subunit of ribonucleotide reductase (RR), the virus is able to replicate only in actively proliferating cells (e.g., tumor cells) with high levels of host RR [39]. In Phase 1 of a clinical trial, G207 showed a strong effect against pediatric high-grade gliomas. Infusion delivery of the virus was carried out through stereotactic intratumoral catheters in concentrations of 10^7^ or 10^8^ plaque-forming units (PFU). Although most patients had large tumors, G207 therapy demonstrated significant antitumor effects, as measured using gadolinium-contrasting MRI scanning. Median overall survival (OS) was 12.2 months against the historically documented 5.6 months. It is worth noting that G207 converted immunologically “cold”—or immunologically “invisible”—tumors to “hot” tumors, with high numbers of infiltrating T-cells and other inflammatory cells detected using immunohistochemistry analysis of pre-treatment and post-treatment tumor tissues [40]. Given the hopeful results from the Phase 1, the Phase 2 clinical trial of G207 in pediatric high-grade glioma is forthcoming (NCT04482933).

The most striking example of an OV based on HSV-1 against glioblastoma is G47∆. G47∆, or teserpaturev, is a third-generation recombinant oncolytic HSV that has a third mutation (*α47* deletion) to its predecessor virus G207. *α47* enables the virus to evade immune surveillance by suppressing the major histocompatibility complex class I (MHC I). Deletion of this gene in G47∆ results in further weakening of the virus in normal cells, but stimulates an antitumor immune response [41]. In Phase 1–2 in patients with progressive glioblastoma, G47∆ showed encouraging outcomes. The virus was administered manually into the one–three target tumor sites repeatedly a maximum of six times (10^9^ PFU each dose). From the second dose, G47∆ was injected into viable areas of the tumor, detected by MRI-contrasting imaging. Median OS from the initial diagnosis (30.5 months) was longer than expected. The authors also note that three patients had long-term survival of more than 46 months. Maximum tolerated dose (MTD), defined as the highest dose of a drug that does not trigger unacceptable adverse events, was not determined for G47∆, since no dose-limiting toxicity was observed in this study. The side effects from G47∆ treatment were related to immune responses to an unnaturally large load of the virus in the located area and were cured with corticosteroids. Thereby, G47∆ showed a survival benefit and a good safety profile, which led to the approval of this recombinant oncolytic virus in Japan [16,42]. To date, G47∆ is the first and is so far the only approved OV for glioblastoma treatment; this encouraging achievement provides an incentive for the development and research of new drugs based on recombinant viruses.

Another interesting illustration of recombinant oncolytic HSV-1 is C134. C134 shows the deletion of the ICP34.5-encoding gene (*γ34.5*), providing lysis of proliferating glioblastoma cells. An additional antitumor response is accomplished through danger signals released from lysed tumor cells and subsequent activation of cytotoxic T-lymphocyte (CTL) response [43]. C134 is also characterized by the presence of an insertion of the protein kinase R (PKR) evasion gene IRS1 from an evolutionary distant cytomegalovirus HCMV, enabling C134 to multiply much more effectively in tumor cells without triggering toxicity [22]. The I phase clinical trial of C134 against recurrent glioblastoma is currently underway (NCT03657576).

### 2.2. Adenovirus

Adenoviruses are non-enveloped double-stranded DNA viruses with a broad range of hosts, including humans [44]. Adenoviruses are the preferred variants for oncolytic virotherapy due to their ability effectively take over the machinery of the host cell [45]. DNX-2401 (Delta24-RGD, tasadenoturev) is a tumor-selective, replication-competent oncolytic virus based on adenovirus type 5. DNX-2401 was designed with 24 base pair deletion in the retinoblastoma (Rb)-binding domain of the *E1A* gene. The product of this gene, mutant E1A (mE1A), normally binds Rb protein to release the transcription factor E2F, that promotes modulation into the S-phase of cell cycle; in turn, this enhances viral replication. Since the Rb pathway is disrupted in glioblastoma cells, E2F is in a free state and the virus can replicate. However, replication in healthy cells is limited due to the inability of the virus mE1A to bind the Rb protein [46]. It is known that the human adenovirus type 5 binds to coxsackievirus and adenovirus receptor (CAR), which has low expression on the surface of glioma cells, resulting in poor infectivity of DNX-2401 for this tumor type. To overcome this problem, the RGD-peptide (arginine-glycine-aspartate)—with high affinity to integrins αvβ3 and αvβ5, which are widely present on the surface of glioblastoma cells—was introduced into the fiber knob virus receptor [47]. Thus, the described genetic modifications specifically target the virus to the tumor cells and contribute to its replication in glioblastomas.

Phase 1 of a clinical trial (NCT01582516) of Delta24-RGD using convection-enhanced delivery (CED) did not achieve a significant increase in the life expectancy of patients. The median OS was 129 days for the group of 20 patients with recurrent glioblastoma. However, six patients had an OS of more than 6 months, including long-term survival of 2.5 and 7.5 years. Four patients demonstrated tumor response on MRI; one of them underwent complete regression and was alive after 8 years. Immunohistochemistry analysis of resected treated tumors revealed the presence of intratumoral CD8+ and T-bet+ T-cells after Delta24-RGD treatment, suggesting an immunogenic antitumor response [48].

Another promising adenovirus being tested against glioblastoma is NSC-CRAd-S-pk7. The native E1A promoter has been replaced with a survivin promoter to avoid non-specific viral replication and enhance the oncolytic effect in malignant glioma [49]. Survivin is a member of the inhibitor of apoptosis (IAP) family, characterized by overexpression in glioblastoma cells and associated with poor prognosis [50]. Thus, the adenoviral vector with a survivin promoter is a prospective drug; it works by targeting transcription in glioma cells. Also, Ad5 fiber protein was modified through the insertion of a polylysine sequence (pk7) [49], Pk7, which naturally binds to heparin sulfate proteoglycans—these are overexpressed in glioma. Neural stem cells (NSCs), as a cell carrier of recombinant adenovirus, enhance a selective glioma through aiming for NSC-CRAd-S-pk7. NSCs are multipotent progenitor cells, which can be engineered to express a variety of anticancer agents, including OVs; due to its inherent tumor-tropic properties, it can be exploited for targeted delivery [51]. A Phase 1 trial (NCT03072134) showed that NSC-CRAd-S-pk7 was safe and exhibited a median OS of 18 months for patients with newly diagnosed high-grade glioma. After the surgical resection of the tumor, the virus was injected in up to ten sites of the tumor bed. The virus did not adversely affect the quality of life of the patients in the trial. Also, using flow cytometry and immunohistochemistry analysis, it has been shown that the recombinant adenovirus influences host immunity, with increased CD8+/CD4+ T-cell ratios [49]. A clinical trial of NSC-CRAd-S-pk7 against recurrent high-grade glioma is currently underway (NCT05139056).

### 2.3. Poliovirus

Polioviruses belong to the Picornaviridae family. They are small, non-enveloped, single-stranded RNA viruses [52]. The poliovirus naturally binds to CD155, also known as poliovirus receptor (PVR), which is expressed at high levels in many tumor cells, including glioblastoma cells [53]. Based on this affinity, an oncolytic virus, PVSRIPO, was developed. PVSRIPO is the poliovirus type 1 (Sabin) vaccine containing a heterologous internal ribosomal entry site (IRES) of human rhinovirus type 2 instead of the poliovirus IRES [54]. IRES is a nucleotide sequence that promotes ribosome assembly and initiates translation by recruiting different translation factors under stress conditions like viral infection [55]. Therefore, the change in PVSRIPO IRES permits this virus to replicate in tumor cells, avoiding replication in normal neurons. Moreover, the weakening of IRES leads to the inability of the PVSRIPO mRNA to recruit the eukaryotic initiation factor 4G (eIF4G), preventing the translation of the virus protein in normal cells and providing a cytotoxic effect in tumor tissue only [56].

In Phase 1 of a clinical trial (NCT01491893), PVSRIPO showed encouraging results for 61 patients with recurrent grade IV malignant glioma. Intratumoral infusion of the virus with dose escalation yielded a high overall survival at 36 months, with the first patients alive more than 6 years post-PVSRIPO treatment [57]. PVSRIPO did not promote systemic autoimmune reactions. However, patients suffered from peritumoral inflammation associated with the location of the tumor, treated with bevacizumab to reduce local edema [58]. At present, PVSRIPO is in Phase 2 of the clinical trial (NCT02986178).

### 2.4. Parvovirus

H-1PV is a small rat protoparvovirus, a member of the Parvoviridae family with linear single-stranded DNA molecules. The cytotoxicity of recombinant parvovirus is attributed to the viral regulatory nonstructural protein NS1, characterized as the major regulator of viral DNA replication and transcription [59]. The natural host of H-1PV is the rat; in humans, it is nonpathogenic. This provides an advantage for using this virus as an OV, since H-1PV may have a wider therapeutic window before the appearance of neutralizing antibodies. In glioma cells, H-1PV can induce lysosome-dependent cell death, with the regression of the two cathepsin inhibitors, cystatin B and C, enabling the virus to overcome the resistance of glioma cells to common cytotoxic agents (e.g., cisplatin) or to soluble death ligands (e.g., TRAIL) [60]. ParvOryx01, created based on the strain H-1PV, was used in a Phase I/IIa clinical trial against primary and recurrent glioblastoma (NCT01301430). The virus was administered intratumorally or intravenously. Nine days after treatment, tumors were resected and ParvOryx01 was re-administered around the resection cavity. H-1PV treatment was safe and well tolerated by patients, and the MTD was not reached. Clinical response was independent of dose or mode of ParvOryx01 administration, and median OS was 464 days for the heterogeneous patient cohort [61].

### 2.5. Other Oncolytic Viruses

In addition to the listed OVs that have reached clinical trials for glioblastoma, there are a number of studies with oncolytic viruses that provide hopeful results.

Vaccinia virus (VACV) is an Orthopoxvirus from the *Poxviridae* family. Orthopoxviruses have a double-stranded DNA genome and a cytoplasmic replication cycle with production of two forms of infectious virus particles called wrapped virion (WV) and extracellular enveloped virion (EV). EVs are responsible for early virus dissemination and WVs are released after cell lysis to mediate further viral infection [62]. VACV has no defined receptor for vaccinia uptake and naturally infects almost all cell types; this makes this virus a reliable basis for designing an OV [63].

TG6002 is a Copenhagen strain of VACV with two deleted genes: thymidine kinase (*TK*) and a subunit of ribonucleotide reductase (*RR*). *TK/RR*-deleted VACV replication depends on the cellular TK and RR, which is known to be overexpressed in tumor cells. The double deletion makes TG6002 safe for healthy cells [64]. TG6002 is armed with the suicide gene *FCU1*, encoding a bifunctional chimeric protein that catalyzes the conversion of nontoxic 5-fluorocetosine (5-FC) and 5-fluorouridine monophosphate. The expression of the *FCU1* gene by the virus makes it a possible candidate for targeted chemotherapy within the tumor [65]. In preclinical trials, TG6002 showed a convincing antitumor effect in multiple human xenografts, including glioblastoma [31].

One more recombinant vaccinia virus is VV-GMCSF-Lact, which was designed based on a Russian L-IVP strain of the VACV. The virus contains deletions of viral thymidine kinase (*TK*) and growth factor (*VGF*) fragments, providing highly selective replication in tumor cells, evading normal cells [66]. Recombinant VACV also has insertions of *GM-CSF* and oncotoxic protein lactaptin genes. Lactaptin is a fragment of human milk kappa-casein (residues 57–134), and it has been found to induce the apoptotic death of various cancer cell cultures [67,68]. GM-CSF is a granulocyte–macrophage colony-stimulating factor that evokes an antitumor immune response. These modifications make VV-GMCSF-Lact a strong antitumor agent for the treatment of malignant neoplasms, including high-grade glioma. Investigating the therapeutic efficacy of the virus against glioblastoma, researchers have evidenced VV-GMCSF-Lact’s ability to suppress the growth of glioma cells both in vitro (immortalized and patient-derived cell cultures) and in vivo (cell-line-derived and patient-derived xenografts) models. The ability of VV-GMCSF-Lact to cross the blood–brain barrier (BBB) and selectively replicate in glioma cells was also demonstrated, making VV-GMCSF-Lact a promising drug for further clinical trials [32].

Myxoma virus (MYXM) is a pathogen that is only dangerous for European rabbits and does not facilitate disease in humans [69]. MYXM, like the VACV, belongs to the *Poxviridae* family and does not require specific receptors for cell entry [70]. Along with the natural oncolytic activity, the above properties make a MYXM a suitable candidate for virotherapy. Indeed, MYXM shows high efficiency in eliminating tumors in mice with gliomas and is a promising drug for antitumor therapy [34,35,36].

The *Paramyxoviridae* family, including the measles virus (MV), the Newcastle disease virus (NDV), and other viruses inducing infections in respiratory tract, is characterized by single-stranded RNA molecules. MV is a human pathogen that often affects young children. A live attenuated vaccine designed to protect humans against measles has been shown to also have an oncolytic effect against various types of malignant neoplasms [71]. The preferred targets for virus cell entry are receptors CD46 and nectin-4 that are overexpressed on the surface of tumor cells, making the MV a possible weapon against tumors [72,73,74,75,76,77,78,79]. Numerous preclinical studies of MV (MV-CEA) with a soluble component of the human carcinoembryonic antigen (CEA), serving as a traceable marker peptide, have been shown to have significant antitumor effects against glioblastoma xenografts [80,81,82,83]. NDV is an avian virus, which is nonpathogenic for humans, with natural oncolytic and immunostimulatory properties [84]. Thus, NDV is a promising basis for the design of a therapeutic agent with high efficacy against tumors, including glioblastoma. Various trials of NDV have shown promising results, with inhibition of tumor growth and effects of prolonging the survival of glioma-bearing mice [85,86,87].

Thus, different recombinant viruses show impressive results when used as monotherapies against high-grade gliomas. However, despite the obvious advantages of virotherapy, there are certain limitations caused by the nature of the viral agents. One of the main problems is the likelihood of a decrease in therapeutic efficacy due to the formation of antiviral neutralizing antibodies. For example, in preclinical studies of the reovirus, its intravenous administration to immunocompetent tumor-bearing mice initially inhibited tumor growth, but 3 weeks later, due to an increase in anti-reovirus antibody levels, tumor growth resumed. Therefore, the introduction of large doses of the viral agent before the possible appearance of neutralizing antibodies is recommended, in addition to the use of the virus in combination with antitumor chemotherapy [88]. Also, utilizing OVs as monotherapies does not always lead to achieving a maximum therapeutic effect or the induction of sufficient oncolysis to create long-term adaptive antitumor immunity [89]. Thereby, to expand antitumor efficacy, oncolytic virotherapy is often exploited in combination with various therapeutic schemes. The most interesting and promising examples of combination therapy against glioblastoma will be considered below.

## 3. Oncolytic Viruses Armed with Transgenes

To enhance the therapeutic efficacy of virotherapy, OVs are armed with transgenic insertions that can both modulate the immune response against the tumor and induce or enhance of tumor cell death.

### 3.1. Oncolytic Viruses with Immunotherapeutic Modulators

It is known that OVs alone can modulate immunosuppressive TME and turn “cold” tumors into “hot” ones [13,90]. Immunoactivating cytokines, like interleukines, are also able to modulate the TME. However, they trigger serious side effects when administered systemically [91]. In particular, the construction of OVs carrying genes for immunostimulating molecules (e.g., IL-12, IL-15, IL-21, GM-CSF) is common.

Interleukin-12 (IL-12) enhances innate and adaptive immunity, has a direct antitumor effect by suppressing tumor angiogenesis [92,93], stimulates the production of interferon-γ (IFN-γ) [94,95], and is a key regulator of cell-mediated immunity [96]. Thus, IL-12 is of interest for its use in glioma therapy. As mentioned earlier, systemic administration induces toxic effects, so some OVs have been designed with an IL-12 insertion. One of them is M032, a second-generation oncolytic HSV with both deleted copies of *γ34.5* and insertion of the human *IL-12* gene. In a study of M032 for canine sporadic gliomas, OVs have been shown to make a possible contribution to prolonged survival in combination with surgical resection [97]. M002, a variant of M032, which expresses murine IL-12, demonstrated significant antitumor activity in preclinical studies on immunocompetent intracranial mice models of brain tumors compared to three other oncolytic HSV-1s, including G207 [98]. The safety and biodistribution of intracranial administration of M002 and M032 in nonhuman primates was evaluated in [99]; based on the data obtained, M032 entered the first phase of clinical trials for use against recurrent malignant gliomas (NCT02062827). Another oncolytic HSV-1 with IL-12 expression is G47∆-mIL12. G47∆, armed with murine IL-12, showed improved efficacy in immunocompetent glioma-bearing mice through inducing IFN-γ production, downregulating proangiogenic vascular endothelial growth factor (VEGF), and ameliorating the immunosuppressive TME [100].

Interleukin-15 (IL-15) plays a major role in the development, survival, and functioning of NK and T cells [101,102]. IL-15 can boost antitumor response when expressed from the OVs. In vivo, oncolytic HSV-1 OV-IL15C, expressing human IL-15 and its receptor IL-15Rα, were demonstrated to have significant suppression of tumor growth and prolonged survival in glioma-bearing mice compared to a therapy using the same OV without *IL-15/IL-15Rα* gene insertion. The IL-15–IL-15Rα complex did not facilitate severe systemic inflammation, but induced antitumor immunity locally due to the local administration of the virus [103].

Interleukin-21 (IL-21) regulates innate and adaptive immunity through controlling the proliferation and function of CD4+ and CD8+ T cells, inducing NK cell activity and suppressing the activity of Treg cells [104]. Vaccinia virus rTTV∆TK-mIL21 with the *IL-21* insertion and expressing the corresponding protein IL-21 has been shown to have an antitumor activity against murine glioma GL261 and has also been demonstrated to provide a benefit through modulating the TME by raising the abundance of NK cells and CD4+ and CD8+ T cells while reducing inhibitory Treg cells [105].

Granulocyte–macrophage colony-stimulating factor (GM-CSF) is responsible for the induction of granulocyte differentiation and macrophages and dendritic cell maturation. GM-CSF also has an ability to induce humoral and cell-mediated immune responses, including increasing the immunogenicity of tumors [106]. Therefore, it is used as immunoactivating transgene in the genome of OVs. The most illustrative examples of the therapeutic effect of expressing GM-CSF in combination with OVs action are T-Vec and JX-594. T-Vec, or Talimogene Laherparepvec, is a double-mutated HSV-1 with insertion of human *GM-CSF* gene into the *γ34.5* loci [107]. T-Vec is the first OV approved by the FDA for the treatment of unresectable melanoma. The insertion of *GM-CSF* gene results in local cytokine production and subsequent systemic antitumor response [108]. Pexa-Vec, or JX-594, is a vaccinia virus with *TK* deletion and insertion of *GM-CSF* and *lacZ* genes. Trials on patients with hepatocellular carcinoma showed high antitumor efficacy of the drug even with systemic administration with concomitant dose-dependent expression of GM-CSF [109,110,111,112]. Another oncolytic vaccinia virus, VV-GMCSF-Lact, with the insertion of *GM-CSF* gene, was shown to hold a high level of this protein in the culture medium of tumor cells infected with recombinants VV-GMCSF-Lact and VV-GMCSF-dGF (the same virus without lactaptin gene insertion); meanwhile, no GM-CSF was detected in the same conditions with the wild-type virus L-IVP [113]. Investigation of VV-GMCSF-Lact also confirms the enhancement of the antitumor effect of recombinant viruses with *GM-CSF* insertion against different types of malignancies, including glioblastoma [32].

### 3.2. Oncolytic Viruses with Apoptotic Inducers

Tumors are characterized by impaired signaling pathways leading to apoptosis, explaining resistance to anticancer therapy. For example, impaired functioning of the PI3K/Akt signaling pathway is commonly observed in glioblastomas [114]. Activated AKT kinase induces overexpression of the *MGMT* proto-oncogene, a negative regulator of the p53 protein, mediating its degradation [115]. Thus, another way to improve the therapeutic effect of virotherapy is the arming of OVs with pro-apoptotic transgenes that can additionally induce the apoptotic death of tumor cells.

The most famous apoptosis inducer protein is p53, which plays numerous biological roles, including cell differentiation, cell cycle arrest, DNA repair, and of course apoptosis [116]. An abnormal p53 is one of the signs of glioblastoma, affecting a more unfavorable prognosis of the disease [117,118]; this is why p53 is of great interest to researchers, including as a transgene for OVs to enhance the oncolytic potency. One such virus is rNDV-p53, a recombinant Newcastle disease virus that expresses p53. In vivo studies have shown that the virus inhibits the growth of glioma and prolongs the survival rate of tumor-bearing mice compared to the rNDV and p53 alone. It has also been demonstrated that rNDV-p53 induces apoptosis of glioma cells by activating genes associated with apoptosis, while increasing the number of apoptotic bodies [119]. Adenoviral vector BB-102 with insertions of wild-type *p53*, *GM-CSF* and *B7-1* (immune co-stimulatory molecule) genes significantly inhibited tumor growth in in vivo experiments compared to the control adenovirus Ad-GFP expressing the fluorescent green protein GFP [120].

Tumor-necrosis-factor-related apoptosis-inducing ligand (TRAIL), a member of a superfamily tumor necrosis factor (TNF), can induce apoptosis of different types of tumor cells; this is because TRAIL death receptors are highly expressed on the surface of oncotransformed cells, but are absent on normal cells [121,122,123]. Being expressed by OVs, TRAIL may contribute to tumor growth inhibition. For example, the Newcastle disease virus Anhinga strain (NDV/Anh), expressing a soluble TRAIL protein (NDV/Anh-TRAIL), was shown to have a greater antitumor efficacy compared to wild-type NDV/Anh and control group of mice with subcutaneously transplanted gliomas [87]. In the work published by Liu Y. et al., an adenovirus (AdsTRAIL) expressing an untagged soluble TRAIL protein was studied in orthotopic xenograft models of glioblastoma. Intratumoral injections of the virus led to a decrease in tumor growth and prolonged survival compared to the control group [124,125]. In another study of adenovirus (Ad.hTRAIL) with *TRAIL* gene insertion, intratumoral virus administration suppressed the growth of human glioblastoma xenografts in murine models [126,127].

OVs armed with transgenes of various actions are becoming a more preferred therapeutic agent due to their increased spectrum of action. In addition to the direct lysis of tumor cells, determined specifically by the natural life cycle of the virus, the expression of transgenes makes it possible to locally induce both an immune response and tumor cell death.

Proceeding from the low effectiveness of monotherapy against glioblastoma, combination therapy options based on virotherapy are currently being developed and used in multiple clinical trials. Next, we will consider the most notable examples of the application of a combination of therapeutic approaches for glioma (Figure 2).

## 4. Combination of Oncolytic Virotherapy and Current Standard of Care for Glioblastoma

Currently, there is an accepted standard for the treatment of glioblastoma, including maximal surgical resection of the tumor followed by radio- and chemotherapy. This scheme ensures the survival of patients with this diagnosis, on average, up to 15 months from the start of the treatment; thus, there is an ongoing need to develop more effective therapy regimens.

### 4.1. Virotherapy in Combination with Resection of the Tumor

Resection of the tumor is the main method of dealing with gliomas, enabling the maximal removal of the neoplasm. However, the resection of a brain tumor is a highly individualized method based on tumor size and shape, location in vital areas of the brain, and surrounding blood vessels [128]. Moreover, glioblastomas are characterized by a high invasive potential, in which tumor cells migrate into the surrounding healthy parenchyma, contributing to invariable recurrence and incurability of the tumor even in the case of maximal surgical resection [5]. To overcome this limitation, fluorescent compounds have been used that selectively accumulate in tumors and induce a fluorescent spectrum after exposure to light of a certain wavelength. This approach allows practitioners to distinguish between tumor cells and perform a more accurate resection. The most promising compound for fluorescent-guided surgery is 5-aminolevulinic acid (5-ALA), approved by the FDA in 2017. 5-ALA is metabolized in the mitochondria of glioma cells with the formation of protoporphyrin IX, absorbing blue light (375–440 nm). Accumulation of 5-ALA in tumor cells permits the better visualization of tumor margins and peritumoral sites compared to contrast-enhanced MRI, leading to longer recurrence-free survival of patients [129].

Another way of combating migrating glioblastoma cells is through treating the post-resection cavity with an OV that can selectively infect and replicate in those tumor cells that have escaped surgery. Adenovirus NSC-CRAd-S-pk7 [49], herpes simplex virus G207 [17], and parvovirus ParvOryx01 [61] have already been tested with this application in clinical trials. This option ensures the elimination of individual tumor cells without arousing strong toxic effects from massive cell death in the whole tumor and reduces the risk of glioblastoma recurrence.

### 4.2. Virotherapy in Combination with Chemotherapy

Chemotherapy with temozolomide (TMZ) is currently a gold standard for the treatment of glioblastoma, approved by the FDA in 2005. Temozolomide is an alkylating agent with a cytotoxic activity against tumor cells [130,131]. However, glioblastoma is resistant to this type of treatment due to the cell-repair systems that are capable of repairing double-stranded or single-stranded DNA breaks caused by alkylating individual nucleotide bases, ultimately making the therapy ineffective [132]. The main stumbling block for chemotherapy is O6-methylguanine-DNA-methyltransferase (MGMT), one of DNA’s repair enzymes. When the *MGMT* promoter is methylated, the activity of the enzyme also decreases; however, with normal *MGMT* expression, the enzyme restores damage in tumor cells evoked by alkylating agents [133,134]. One of the options for improving the therapeutic effect of chemotherapy is its combination with oncolytic virus therapy.

In a Phase 1 study with patients with high-grade malignant gliomas of the NSC-CRAd-S-pk7 virus, standard radiotherapy and chemotherapy were started 10–14 days after resection and virus injection [49]. The authors report no significant toxic effects related to virotherapy. In addition, the disease has been shown to have stabilized in most patients following this treatment. Also, in a subgroup with unmethylated *MGMT* promoter gliomas, the median progression-free survival was 8.8 months, and the median overall survival was 18.0 months. For patients with methylated *MGMT* promoter tumors, progression-free survival and overall survival were 24.2 months and 36.4 months, respectively. The modulation of the immune response, particularly an increase in the population of CD8+ T-lymphocytes, has also been demonstrated.

A clinical trial of DNX-2401 virus was performed with patients with recurrent gliomas [21]. After intraparenchymal administration of the virus, a course of four cycles of 150 mg/m2 temozolomide was administered 2–4 weeks later. It was shown that such therapy prolonged the life expectancy of patients. In addition, higher fibroblast growth factor 2 (FGF2) expression was correlated with better survival. FGF2 is known to increase the sensitivity of tumor cells to viral therapies [135]. Also, IFN-γ expression was lower in patients with better survival. The data obtained by the authors have led to additional studies of the mechanisms that determine efficiency.

Delta24-RGD has synergistic cytotoxic effects along with TMZ [136]. Moreover, authors have reported the activation of apoptosis by examining the induction of Caspases-3 and Caspases-7. In addition, autophagy processes are activated. In an in vivo experiment, the authors showed a decrease in the influx of NK cells, dendritic cells, and CD8+ T cells in the tumor when TMZ was used before virus therapy. However, the administration of Delta24-RGD first and then TMZ leads to an increase in CD8+ population. A Phase 1 clinical trial using a combination of Delta24-RGD with TMZ in human glioblastoma has already been completed, but the results have not published yet (NCT01956734).

The oncolytic adenovirus vector Ad-Delo3-RGD is characterized by Y-box protein (YB-1)-dependent replication, determining the selectivity for tumor cells [137,138]. YB-1 influences the development of multidrug resistance and is a marker for predicting treatment outcome [139]. In the combination of this virus with TMZ in vitro, there is a decrease in the viability of glioblastoma cells [23,139]. On the xenograft model, a significant reduction in tumor volumes is observed compared with monotherapy. Moreover, both in monotherapy and in combination therapies, the virus has been shown to induce the apoptosis of tumor cells. Another oncolytic adenovirus, ICOVIR-5, also has an enhanced effect in combination with TMZ both in vitro and in vivo [24,140].

In preclinical studies, the only oncolytic virus approved for use in patients with glioblastoma, G47Δ, was evaluated in combination with etoposide [18,39]. Etoposide is a semi-synthetic derivative of podophyllotoxin. It inhibits the relegation of cleaved DNA molecules by topoisomerase II, ultimately leading to late S and G2 cell cycle arrest. Under the influence of this combination, the viability of glioblastoma cells obtained from tumor samples of patients decreased. In addition, the induction of apoptosis was shown, as indicated by an increase in the level of cleaved Caspase-3. Moreover, the life expectancy of tumor-bearing mice treated with this combination was higher compared to that of the monotherapy and control groups. G47Δ has also been studied in combination with temozolomide [19,39]. The joint action is characterized by synergy both in vitro and in vivo.

The myxoma virus (MYXV) M011L protein inhibits apoptosis by interfering with cytochrome c release. The potential oncolytic MYXV construct, lacking M011L (vMyx-M011L-KO), induces apoptosis in brain-tumor-initiating cells (BTICs) that were obtained from glioma surgical samples [36]. In addition, when immunocompetent mice were treated with transplanted BTICs, there was prolonged survival, enhanced in combination with TMZ.

Another MYXV was investigated in a human glioblastoma model [141]. In combination with radiotherapy and/or temozolomide, the virus significantly reduces the viability of U118 cells. The authors observed an increase in the level of cleaved Caspase-3 and a decrease in the phosphorylation of protein kinase AKT in cells when treated with MYXV in conjunction with radiation and chemotherapy. This indicates a greater activation of apoptosis in combination therapy than in monotherapy with the studied techniques.

In a Newcastle disease virus (NDV) study, together with TMZ in vitro, a synergistic effect was shown, as well as the inhibition of AKT and protein kinase AMPK, indicating the induction of apoptosis. In addition, this work reported a prolongation in life expectancy and a decrease in tumor volumes with the combined use for tumor-bearing rats [142].

Thus, the combination of chemotherapy with OVs creates a more effective treatment regimen for patients with glioblastoma due to supplementary induction of tumor cell apoptosis. Furthermore, this method allows the tumor to be influenced through different mechanisms, leading to an improved therapeutic effect.

### 4.3. Virotherapy in Combination with Radiotherapy

Radiotherapy has been playing a major role in glioma therapy in recent decades. For low-grade gliomas, the preferred dose is 50.4 Gy [143,144]. At the same time, in patients with low-grade gliomas who received early radiation therapy, the time to progression of the disease is longer compared to people who are under observation and suffer from delayed radiation therapy [145]. For patients with high-grade glioma who are receiving radiotherapy for the first time, a total dose of 60 Gy is recommended. The median time to relapse is 4.3 years. However, due to the development of glioma resistance to radiotherapy, the rationality of irradiation remains questionable. Thus, the use of radiotherapy with oncolytic viruses can enhance the effect and improve quality of life for patients.

At the moment, a limited number of preclinical and clinical trials of OVs with radiotherapy have been carried out. One of the first works on this topic was published by J D Bradley et al. [146,147]. The authors showed that genetically modified HSV R3616 was synergistic with radiotherapy; the authors observed both a significant reduction in tumor volumes and a prolongation in the lifespan of animals with intracranial U87 MG tumors. Encouraging results have been obtained in the G207 clinical trial, where patients who were 7–18 years of age with high-grade glioma received a dose of 5 Gy after virotherapy. The therapy has been shown to have low toxicity. In addition, during treatment with an oncolytic virus, the number of tumor-infiltrating cells of the immune response rose. Also, there was an increase in median overall survival [40].

Another study with pediatric patients used DNX-2401 virotherapy followed by radiotherapy [148]. The group of patients with diffuse intrinsic pontine glioma (DIPG) was characterized by a reduction in tumor volumes and an increase in median overall survival. The authors also note that, by the time the paper was published in late 2022, one of the patients had been progression-free for 38 months.

Ad5-Delta24RGD has also been shown to have promising results in in vivo experiments on glioma xenografts [25,149]. In addition to the fact that all animals showed complete tumor regression with a combination of virotherapy and radiotherapy, the authors showed that such a combination can significantly reduce the required effective dose of the virus. This finding is extremely important, since lowering the dose will reduce the manifestation of possible adverse events.

An interesting work was published by B. Geoerger et al., which investigated the combination of ONYX-015 and radiotherapy [26,150]. In this work, patient-derived p53-mutant and p53-wild-type xenografts were used, bringing the research results closer to those which might be derived from real primary tumors. Animals received irradiation at a dose of 5 Gy followed by 5 days of virus injection. Both models show an additive effect, with the p53 wild-type tumor being characterized by a potentiated effect. In another study, the authors explore the mechanisms responsible for this observation. Kong and colleagues report that, during the ONYX-015 virus infection, the functional activity of p53 is induced in tumors with wild-type p53. In addition, wild-type p53 cells are known to be more sensitive to radiotherapy [151].

In the adenovirus-based vector AdV-tk clinical trial, patients with malignant gliomas were injected with the virus after resection, and radiotherapy was started 7 days later [27]. Such therapy was safe and potentially prolonged survival. In addition, the authors showed an expansion of CD8+ T-lymphocytes population.

For measles virus derivatives that have been engineered to overexpress the human carcinoembryonic antigen (MV-CEA) in in vitro and in vivo models, a synergistic effect of combination with radiotherapy has been shown [152,153]. In addition, the authors showed that there is an increase in apoptosis along the external pathway. The latter fact can be very important for the treatment of gliomas, since the development of necrosis, and, accordingly, inflammation is an undesirable side effect.

Thus, various studies demonstrate that virotherapy in combination with radiotherapy may be a promising approach in the treatment of gliomas. Viruses can promote radiation-mediated apoptosis, enhancing the therapeutic effect. In addition, this multimodal treatment regimen can reduce the risk of recurrence and will significantly improve quality of life for patients.

## 5. Combination of Oncolytic Virotherapy and Immunotherapy for Glioblastoma

Glioblastoma is characterized by a highly immunosuppressive TME, explaining the poor response to immunotherapeutic strategies. However, infection of tumor cells with OVs activates an antitumor immunity and transforms an immunologically “cold” tumors into a “hot” ones [13]. Lysis of tumor cells under the influence of OVs stimulates the immune system, arousing immunogenic cell death. This pathway of cell death leads to the release of molecular patterns associated with danger (DAMPs and PAMPs) and signaling molecules of innate immunity (HMGB1 and ADP); in turn, this serves to transform the immunosuppressive TME into a proinflammatory one, thereby activating an innate immunity in nearby cells [154]. Many studies also confirm that virotherapy can enhance the effect of immunotherapy, and the combination of these two methods may open new perspectives and offer hope for the development of an effective regimen for glioblastoma treatment.

### 5.1. Immune Checkpoint Inhibitors

Immune checkpoints (ICs) are regulatory signaling pathways involved in immune modulation by influencing T-cell abundance [155]. The most common immune checkpoints include PD-1/PD-L1, LAG-3, and TIM-3. PD-1 is a molecule of the programmed cell death (PD) pathway, involved in inhibiting the activity of cytotoxic T-cells within the tumor. To stimulate the immune response, PD-1 or PD-L1 can be inhibited to block interactions between receptor (PD-1) and ligand (PD-L1) [156]. LAG-3 (lymphocyte activation gene product 3) is a T-cell activation inhibitory receptor that is found on the surface of immune cells (CD4+, CD8+) [157]. TIM-3 (HAVCR2, T-cell immunoglobulin, and mucin domain-containing protein 3) suppresses lymphocyte viability and function and is also a marker of CD4+ and CD8+ T-cell exhaustion [158].

In the case of malignant neoplasms, immune surveillance is avoided due to incorrect work of immune checkpoints, significantly influencing the effectiveness of various therapeutic approaches, since the interactions between ICs and its ligands negatively alter T-cell function, complicating the physiological response against tumor-associated antigens. In this regard, immune checkpoint inhibitors (ICIs) are now being actively investigated and are showing encouraging results in clinical trials for different tumor types [159,160,161]. However, ICIs monotherapy does not always provide a positive therapeutic effect in preclinical models and clinical studies for glioblastoma, since this type of tumor is characterized by low T-cell infiltration and immunosuppressive TME. For example, in the Phase III clinical trial, the median OS for patients with recurrent glioblastoma was almost the same between the group treated with nivolumab (anti-PD-1) and the group treated with bevacizumab (anti-VEGF), which inhibits tumor growth by targeting neovascularization endotheliocytes, leading to the disruption of the tumor’s blood supply [162]. In this case, the combination with OVs may improve the therapy of glioblastoma by raising the infiltration of immune cells into tumors, making them more susceptible to immunotherapeutic drugs. It has also been shown that OVs are able to improve the therapeutic effect of ICIs by increasing the expression of PD-L1 through the induction of the release of interferons from virus-infected tumor cells into the TME [163].

A study of oncolytic measles virus (MV) and anti-PD-1 therapy against murine glioma GL261 in immunocompetent C57Bl/6 mice has been conducted; in this multimodal therapeutic regimen was shown to synergistically enhance the survival of mice and increased antitumor activity was observed in comparison with single-agent therapies. An increased T-cellular influx into the mice brains was noted using T-cell MRI scanning and FACS analysis [164].

The oncolytic HSV G47∆ with murine interleukin-12 insertion (G47∆-mIL12) has also been investigated in combination with anti-CTLA-4 and anti-PD-1 antibodies on CT-2A and 005 GSC-derived gliomas in C57Bl/6 mice. Triple therapy of 005 GSCs resulted in 89% long-term survivors compared with dual checkpoint inhibitors alone (37%). Triple combination treatment of CT-2A glioma, which is more aggressive than 005 GSC-derived tumors, led to half of the mice experiencing long-term survival. Moreover, checkpoint inhibitor therapy did not affect the virus spread and did not promote its toxicity. Both groups of the surviving mice underwent repeated intracranial transplantation on day 183 and 109, respectively, after initial cell transplantation; a 5-fold increased number of original tumor cells was grafted in both groups, and this did not lead to tumor formation. This result indicates the presence of immunological memory after multiple therapeutic interventions [20].

Another recombinant virus, used in combination with ICIs, is VVL∆TK-STC∆N1L-mIL21. It is an oncolytic vaccinia virus with deleted TK and N1L genes that provides tumor selectivity; additionally, N1L deletion induces an antitumor immune response. Combined therapy of VVL∆TK-STC∆N1L-mIL21 with anti-PD-1 antibody was demonstrated to have a strong therapeutic effect against murine glioma GL261, with 80% of the C57Bl/6 mice being cured, and no recurrence occurred in the 180-day observation period. The induction of the polarization of M1 macrophages was noted in all groups receiving OVs, reflecting the ability of the virus to remodel the TME. The highest polarization of macrophages was shown in the combined-therapy-treated group. Increased levels of CD8+ memory T cells was observed in the group treated with VVL∆TK-STC∆N1L-mIL21 and anti-PD-1 compared with the control group and the other experimental groups; these received only virus monotherapy [33].

A study of the oncolytic poliovirus PVSRIPO in models of immunocompetent glioma in combination with PD-1/PD-L1 blockade revealed the effects of the therapy on glioma-associated macrophages and microglia, contributing to immune suppression and tumor progression. The combination therapy resulted in microglial activation, suppression of tumor growth, and durable remissions [30]. Since combined immune-viral therapy is fairly new approach, most of the evidence comes from preclinical trials; however, clinical trials are beginning to emerge for combination therapies including OVs and ICIs. Thus, Phase II clinical trials of the combination therapy of adenovirus DNX-2401 and pembrolizumab (an immune checkpoint PD-1 inhibitor) have been conducted for recurrent glioblastoma, but the results have not been published yet (NCT02798406).

### 5.2. Adoptive Cell Therapy

Adoptive cell therapy (ACT) aims to suppress the defense of the tumor against immune attack by targeting patient-derived T-cells to recognize and eliminate tumor cells. The greatest success in this direction has chimeric antigen receptor (CAR) T-cell therapy, especially for the treatment of hematological malignancies and some types of solid tumors. The mechanism of targeted cytotoxicity is the recognition by CAR T-cells of specific surface molecules on tumor cells. Several molecules of glioblastoma were identified as potential targets for ACT (e.g., EGFRvIII, IL13Rα2, HER2, B7H3) [165]. In a clinical trial of EGFRvIII-specific CAR T-cells, low efficacy was shown against recurrent glioblastoma. A study of the TME after ACT showed high levels of Treg cells, as well as excessive activation of immunosuppressive molecules [166]. CAR-modified virus-specific T-cells targeting tumor antigens are also being developed. Thus, in the Phase I clinical trials of HER2-specific CAR T-cells, a partial response to the therapy was shown in patients with progressive glioblastoma; the median OS was 11.1 months after the first injection of T-cells and 24.5 months from the moment of diagnosis [167]. Consequently, glioblastomas do not always respond to such treatment, in particular due to low immune cell recruitment and infiltration into the tumor, as well as due to the presence of an immunosuppressive TME and low representation of individual tumor antigens in recurrent glioma.

As previously mentioned, OVs have been shown to be potent inducers of an antitumor immune response, making them suitable companions for synergistic action in the treatment of glioblastoma with ACT. In the combination study of oncolytic adenovirus armed with murine chemokine CXCL11 and B7H3 CAR T-cells, it was shown that CXCL11-oAd improves the recruitment of CAR T-cells into the tumor sites and reprograms the immunosuppressive TME in immunocompetent mice C57Bl/6 with glioma GL261. A TME analysis showed that virus injection leads to high levels of CD8+ cells, NK cells, and M1-macrophages; together, these were shown to improve the therapeutic effect of CAR T-cell therapy [28]. In another preclinical study of oncolytic adenovirus with interleukin-17 insertion (oAd-IL17), in combination with B7H3 CAR T-cells, a significantly improved antitumor efficacy was demonstrated compared with monotherapy of OV or CAR T-cells; the latter brought a limited therapeutic response in xenograft models [29].

### 5.3. Dendritic Cell Vaccines

Dendritic cell vaccines (DCVs) are based on dendritic cells, which are antigen-presenting cells that express MHC class I and II molecules, making them effective endogenous immune cells stimulators [168]. DCVs have been shown to be effective in patients with glioblastoma in various clinical trials (NCT01171469, NCT00766753, NCT00576641, and NCT01280552), but there remains a need to select a therapeutic regimen that provides more complete survival. To improve DCVs’ immunogenicity, using them in combination with OVs can enhance antitumor immunity. Combinational therapy was investigated against different types of malignancies; as for brain tumors, a preclinical study of oncolytic HSV-1 with immature dendritic cells showed prolonged survival and a 90% reduction in tumors in murine neuroblastoma models [16,169].

### 5.4. Peptide Vaccines

Peptide vaccines are small immunological peptides that are processed by antigen-presenting cells (APCs) and have an affinity to human leukocyte antigen (HLA) class 1. Cytotoxic T-lymphocytes (CTLs) recognize these peptides on APCs, leading to the activation of CTLs, their migration to the tumor, and the detection of the required peptide-HLA complex on glioma cells; this enables their elimination, resulting in tumor regression [170]. These peptides can be targeted to specific antigens of glioblastoma, such as proteins EGFRvIII, IDH1(R132H), WT1, and survivin. One of these vaccines is HSPPC-96 (heat shock protein peptide complex-96) that comprises heat shock glycoprotein 96kDa, which is overexpressed in brain tumor cells [171]. Such heat shock vaccines are able to induce a CD8+ T-cell response and show preliminary therapeutic effectiveness against glioblastoma in clinical trials (NCT02722512). Many other peptide drugs have been investigated for gliomas both in monotherapy and in combination with other options, like chemotherapy [172,173,174]. However, there is currently no information on studies of the combined action of peptide vaccines and OVs, despite the fact this may be a very promising field for research.

Immunotherapy as an approach for the treatment of malignant neoplasms is currently actively developing, with encouraging results in clinical trials. In combination with the natural ability of OVs to turn immunologically “cold” tumors to “hot” ones, immunotherapeutic methods can achieve significant antitumor effects compared to monotherapies.

## 6. Limitations and Future Directions

Virotherapy as a therapeutic approach for glioblastoma treatment demonstrates significant antitumor efficacy. However, like any other method, it has certain limitations that require additional consideration. One of these limitations lies in the certain toxicity of viral drugs. The most often adverse events manifest in disorders of the gastrointestinal system (vomiting, nausea) and the nervous system (seizure, headache, cerebral edema). For example, a recombinant nonpathogenic polio-rhinovirus chimera, PVSRIPO, caused severe peritumoral inflammation when administered intratumorally in patients with glioblastomas [58]. As one of the possible options for avoiding such reactions, the methods of viral agent delivery to the tumor should be reconsidered, since intense inflammation and brain edema are extremely undesirable for patients with such a torturous disease. Treatment of the post-resection cavity with viral drugs will help bypass the massive destruction of tumor tissue and the accompanying inflammation. A similar approach has already been used in clinical trials of adenovirus NSC-CRAd-S-pk7 [49], herpes simplex virus G207 [17], and parvovirus ParvOryx01 [61]. In cases of unresectable tumors, an approach can be applied to gradually deliver OVs to different areas of the tumor, distributing the doses over time to minimize the number of tumor cells destroyed at the same time, as was described in a study of oncolytic HSV-1 G47Δ [42]. Systemic or intravenous administration of the drug can also minimize the risks of side effects. However, this delivery method has a strong limitation in the formation of virus-neutralizing antibodies significantly reducing the effectiveness of such therapy.

Regrettably, standard glioma therapy, consisting of the maximal surgical resection of the tumor and subsequent chemo- and radiotherapy, is characterized by low effectiveness; this is because glioblastoma unfailingly recurs after such treatments. The problem lies in the fact that the maximal safest surgical resection often does not remove the entire tumor mass due to its location in vital brain areas. Chemo- and radiotherapy, in turn, lose their potency as glioblastoma develops resistance to these treatment methods. Undeniably, additional therapy options are required to overcome these restrictions. As mentioned earlier, virotherapy can be applied in the treatment of post-resection cavities. This method allows practitioners to minimize the toxicity of the treatment and to reduce the risks of tumor recurrence. As for adjuvant drug agents, OVs can induce apoptotic tumor cell death, increasing the impact of different therapeutic approaches.

Nevertheless, most proposed OVs for the treatment of brain malignancies do not show severe toxicity in the majority of patients studied, and all adverse events are successfully treated with corticosteroids, as confirmed by a study of oncolytic HSV-1 G47Δ [42] and other OVs. Minimal toxic effects from the use of OVs also show their advantage in comparison with other therapeutic approaches. The NSC-CRAd-S-pk7 study noted adverse events, but all of them were related to adjuvant chemo- and radiotherapy [49]. Moreover, interleukins, which are useful in inducing immune responses in patients with glioblastoma, are known to facilitate severe toxicity when administered systemically [91]. At the same time, OVs with insertions of interleukin genes make it possible to bypass this limitation and deliver the necessary immunostimulating molecules directly to the tumor without causing undesirable inflammation.

As for immunotherapy, immunosuppressive TME in patients with glioblastoma is a hot topic for modern researchers. Multiple signaling molecules are actively involved in the suppression of patients’ immune responses, leading to a need for a multimodal approach to the issue. One of the promising immunotherapy methods is the use of ICIs; these have been shown to have encouraging results in tumors of different origins. However, in the case of glioblastoma, it has been shown that targeting only one IC is ineffective; the simultaneous inhibition of two or more ICs is considered more frequently [155]. Moreover, ICIs are often ineffective for application against tumors which are “cold" or “invisible” to the immune system. Meanwhile, virotherapy, which is considered as one of the most powerful immunotherapeutic options, is known to turn immunologically “cold” tumors into “hot” tumors, helping to increase the effectiveness of such treatments, which can then affect the tumors in multiple ways.

In addition to the benefits of virotherapy described below, one more of the undoubted advantages of using OVs against glioblastoma is its antitumor effect, regardless of the patients’ *IDH1* and *MGMT* statuses. Thus, in the study of G47Δ, it was shown that *IDH1* and *MGMT* statuses do not affect the median survival of patients treated with G47Δ [42]. In a study of the combination of oncolytic adenovirus NSC-CRAd-S-pk7 with temozolomide, it was demonstrated that, for patients with unmethylated *MGMT* promoter, survival outcomes remain high compared with temozolomide therapy alone [49]. Unmethylated *MGMT* promoter is known to be a predictor of poor disease outcome because it is not amenable to chemotherapy [175]. These studies confirm the advantage of the application of virotherapy in combination with standard glioma therapy, increasing the effectiveness of antitumor activity.

It is worth highlighting that there are different new approaches for combinations with virotherapies that can be explored to enhance the antitumor effect in the treatment of glioblastoma. Since the application of ICIs leads to significant adverse events, manifesting themselves in hematological and neurotoxic ways (nephritis and myocarditis), next-generation ICIs are being actively developed. Next-generation ICIs are small molecules (2F-Fuc, gefitinib, metformin, ediposide, curcumin, etc.) acting on the pathways of IC glycosylation and ubiquitination/degradation [176]. In the near future, OVs can be used in combination therapy with those next-generation ICIs, leading to successful cancer therapies. A promising investigation field is CRISPR/Cas (Clustered Regularly Interspaced Short Palindronic Repeat), which is an impressive modern genome editing technology. One of the options of using this technology is in treating cancer by removing genes and correcting mutations. This technology has also been used for in vivo testing against glioblastoma, but this method faces certain limitations in delivering the therapeutic agent [177]. Viral vectors, such as adenoviral vectors (AdV), adeno-associated viruses (AAV), and lentiviral vectors (LV), are widely used to deliver the CRISPR/Cas9 system due to their high delivery effectiveness [178]. Thus, OVs can also be used as deliverers of the CRISPR/Cas systems to facilitate the transport of this therapeutic agent directly to the tumor, as well as synergistically enhance the therapeutic potency. As previously described, OVs can contain insertions of different oncotoxic proteins, leading to the enhancement of antitumor activity. A study of the protein nesprin-2 demonstrated that its deficiency in tumor cells inhibits key events in the mitochondrial apoptotic pathway, in particular in the activation of pro-apoptotic proteins Bax and Bak [179]. Thus, nesprin-2 is a one of the vital regulators of apoptosis, whose gene insertion into the oncolytic viral genome may be useful for further research. Therefore, OVs can be used as therapeutic weapons, armed with different gene-editing systems, oncotoxic genes, and other immunomodulating molecules to increase the efficacy of antitumor activity.

## 7. Conclusions

Glioblastoma is one of the most aggressive and malignant tumors of the central nervous system. The complexity of the treatment of glioblastoma is due to a number of reasons, including the location of the neoplasm in the vital areas of the brain, the growth of the infiltrative tumor, and cell resistance to traditional antitumor therapy. The main technique in the treatment of gliomas is the standard therapy regimen, including the maximum possible surgical resection of the tumor followed by chemo- and radiotherapy. In addition, numerous therapeutic approaches continue to be developed in an attempt to defeat this diagnosis.

However, for aggressive tumors like glioblastomas, such monotherapies are not able to completely defeat the tumor, since gliomas almost always develop resistance to the therapy. Furthermore, some therapeutic regimens can facilitate serious toxic effects, which are extremely undesirable for brain diseases. Virotherapy is one of the promising tools for the treatment of glioblastoma, carrying a minimal risk of side effects in humans. OVs are capable of infecting tumors through several mechanisms: selective viral replication in tumor cells, direct cell lysis, and induction of a local immune response (Figure 3). Several oncolytic viruses have already been approved for clinical use, including herpes simplex virus G47Δ, which is the first—and, so far, only—viral drug that has been approved for use against glioblastoma. However, virotherapy also faces certain limitations, for example, the presence of virus-neutralizing antibodies that reduce the oncolytic activity of the drug. In this regard, the approaches of both traditional therapies and newer developments are used in combination with OVs to secure and obtain a synergistic antitumor effect aimed at the neoplasm. Numerous studies have shown that the combination of virotherapy with the traditional treatment regimen prolongs the life of patients diagnosed with glioblastoma. Thus, the treatment of the resection cavity with a viral drug makes it possible to eliminate the malignant cells remaining after surgical resection, and the combination of OVs with chemo- and radiotherapy leads to a synergistic effect and increased apoptosis of tumor cells. The induction of apoptotic cell death and the modulation of the immune response are achieved using transgenic insertions into the viral genome, improving the oncolytic potential of virotherapy. Significant results were obtained by studying OVs with different immunotherapeutic methods. Immune checkpoint inhibitors and CAR T-cell therapy together with the natural immunostimulatory ability of OVs significantly raise tumor infiltration by T cells, enhancing the therapeutic response of the tumor to treatment.

Thus, virotherapy in combination with different approaches provides a multimodal treatment regimen for glioblastoma and requires further research if we are to overcome this dire diagnosis. 

## Figures and Tables

**Figure 1 ijms-25-02042-f001:**
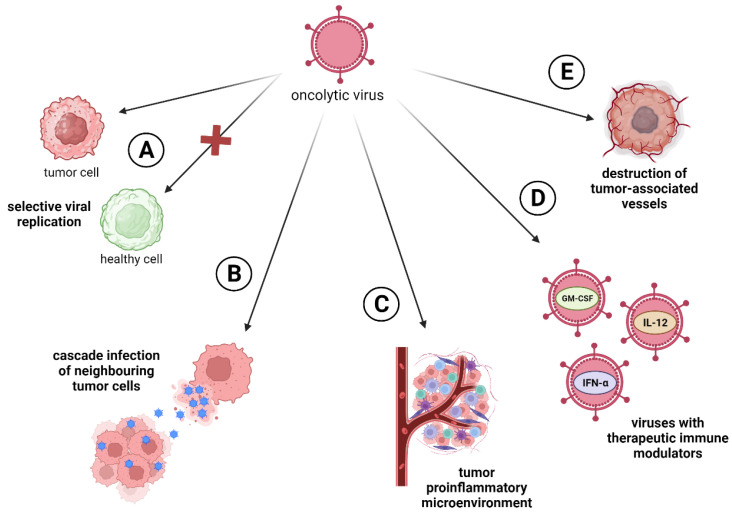
Oncolytic virus’s approaches to targeting tumors. (**A**) Selective viral replication in tumor cells; (**B**) direct cell lysis followed by cascade infection of neighboring tumor cells; (**C**) induction of tumor proinflammatory microenvironment, turning immunologically “cold” tumors into “hot” ones; (**D**) enhancing the oncotoxic ability of OVs through the insertion of therapeutic immune-modulators genes; (**E**) viral infection of tumor-associated cells like endotheliocytes.

**Figure 2 ijms-25-02042-f002:**
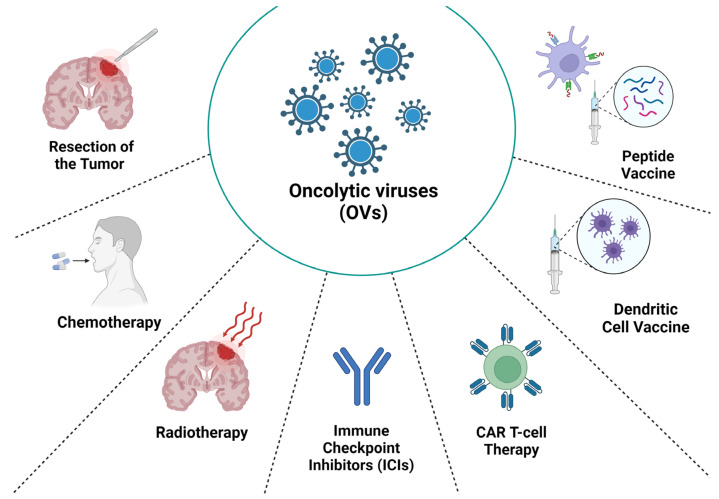
Therapeutic approaches applied in combination with virotherapy to enhance the antitumor effect against glioblastoma.

**Figure 3 ijms-25-02042-f003:**
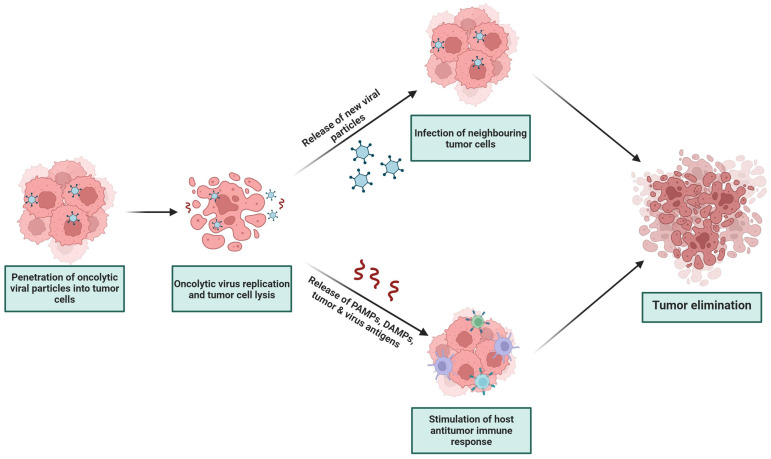
The mechanism of OVs’ action on tumor cell oncolysis.

**Table 1 ijms-25-02042-t001:** Oncolytic viruses applied in monotherapy and in combination with different therapeutic approaches against glioblastoma.

Virus Type	Monotherapy	Resection of the Tumor	Chemotherapy	Radiotherapy	Immune Checkpoint Inhibitors (ICIs)	Adoptive Cell Therapy
Herpes Simplex Virus Type 1 (HSV-1)	G207 (NCT02457845) G47∆ (teserpaturev, DELYTACT) (UMIN-CTR Clinical Trial Registry UMIN000002661) [16] C134 (NCT03657576)	G207 [17]	G47∆ (etoposide), [18], (temozolomide) [19]	G207 (NCT02457845) (NCT04482933)	G47∆ [20]	
Adenovirus	DNX-2401 (Delta24-RGD, tasadenoturev) (NCT00805376) NSC-CRAd-S-pk7 (NCT05139056)	NSC-CRAd-S-pk7 (NCT03072134)	DNX-2401 [21,22] Ad-Delo3-RGD [23] ICOVIR-5 [24] NSC-CRAd-S-pk7 (NCT03072134)	DNX-2401 (NCT03178032) Ad5-Delta24RGD [25] ONYX-015 [26] AdV-tk [27] NSC-CRAd-S-pk7 (NCT03072134)	DNX-2401 (NCT02798406)	CXCL11-oAd [28] oAd-IL17 [29]
Poliovirus	PVSRIPO (NCT02986178)				PVSRIPO [30]	
Parvovirus		ParvOryx01 (NCT01301430)				
Vaccinia virus (VACV)	TG6002 [31] VV-GMCSF-Lact [32]				VVL∆TK-STC∆N1L-mIL21 [33]	vvDD-IL15Rα-YFP [34]
Myxoma virus (MYXM)	vMyxgfp [35]		vMyx-M011L-KO [36]			vMyx-IL15Rα-tdTr [34]

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
