# Peer review of "Combination of Oncolytic Virotherapy with Different Antitumor Approaches against Glioblastoma"

_ijms, 2024, doi:10.3390/ijms25042042_

Round 1
Reviewer 1 Report
Comments and Suggestions for Authors
Ageenko et al described potential methodologies to combine oncolytic viral therapy with other forms of treatment for glioblastoma. They reviewed the current state of investigation for oncologic viral therapy and how such viruses may additionally target other functions and the rationale for combining oncolytic viral therapy with other forms of treatment for glioblastoma. The authors describe experimental models in which oncolytic viruses and various targets including surgery, chemotherapy, radiation therapy, immune checkpoint inhibitors, her T-cell therapy, and then genetic or peptide vaccines may provide additional benefit or even synergistic effects on the tumor.
INTRODUCTION: Adequately describes current standard of care and ongoing investigative therapies for glioblastoma, including oncolytic viral therapies.
REVIEW SECTIONS: Sectioned 2 “virotherapy against glioblastoma” , section 3 “oncolytic viruses armed with trans genes”, section 4 “combination of oncolytic virotherapy and current standard of care for glioblastoma” and section 5 “combination of oncologic virotherapy and immunotherapy for glioblastoma” review the current status of ongoing clinical trials of these forms of therapies alone and in combination with oncolytic viral therapies. This is well researched and contains up to date information regarding the current state of research.
CONCLUSIONS: Section 6 (conclusion) adequately summarizes sections 2-5, with the conclusion that more research is needed.
REFERENCES: All references are appropriate and up to date.
TABLES AND FIGURES: All tables and figures are appropriate and complement/supplement the narrative text.
Author Response
Thank you for your careful review of our manuscript.
Reviewer 2 Report
Comments and Suggestions for Authors
Dear Authors,
Congratulations on your hard work. Here are my recommendations for your manuscript:
- Provide more emphasis on the methodologies employed in the primary studies discussed.
- Highlight the strengths and limitations of the original research to offer a nuanced understanding.
- Include insights into ongoing and future studies to enhance the completeness of the manuscript.
- Discuss potential advancements and emerging trends in virotherapy for glioblastoma.
- Discuss potential challenges and areas where further research is needed.
- Suggest directions that researchers might explore based on current gaps and challenges.
Author Response
Thank you for your careful review of our manuscript. We tried to make changes in accordance with your suggestions. We added therapies and studies during the recruitment phase where possible. In addition, we added a separate paragraph where we highlighted the limitations, advantages and future directions for further research in the area under consideration.Changes in the text are indicated by color.
Reviewer 3 Report
Comments and Suggestions for Authors
Overall, the paper is a substantial contribution to the field, offering a comprehensive overview of current and potential uses of oncolytic viruses in glioblastoma treatment. Further refinement in comparing with non-virotherapy treatments and discussing study limitations would enhance its impact:
- It comprehensively covers recent advances in virotherapy, summarizing findings from multiple studies and clinical trials. However, it lacks a discussion on potential biases or limitations in the studies reviewed.
- The paper effectively compares various oncolytic viruses and their combinations with other therapies. However, it could benefit from a more detailed comparison with non-virotherapy treatments for glioblastoma, to better highlight the advantages and limitations of virotherapy.
- I suggest to make the graph showing the mechanism of action of the viruses on oncolysis of the cells.
- Suggestions for Improvement: Include a section discussing potential limitations or biases in the studies reviewed. Expand on comparisons with alternative glioblastoma treatments to provide a more rounded view of the treatment landscape. A discussion on future directions or emerging research areas in oncolytic virotherapy would be beneficial.
- Grammar errors: 1) line 5: "ParvOryx01 were re-administered around the resection cavity." Is singular. 2) Line 21: "For 3 of 13 patients long-term survival of more than 46 months was observed." Improve clarity, 3) Line 23 "Cytotoxic T-lymphocytes (CTLs) recognize these peptides on APCs leading to the activa- tion of CTLs their migration to the tumor..." Punctuation, 5) Line “29 It worth noting that G207 con- verted immunologically “cold” tumors to “hot”..." Lacks completeness, 6) Line 29: "...immune surveillance is avoided due to incorrect work of immune checkpoints..." Different words should improve clarity, 7) Line 39 "...its DNA do not integrate into host genome..." DNA is singular.
Otherwise a nice paper, worth improvement and reconsidering after revisions.
Comments on the Quality of English Language
Overall, the paper is a substantial contribution to the field, offering a comprehensive overview of current and potential uses of oncolytic viruses in glioblastoma treatment. Further refinement in comparing with non-virotherapy treatments and discussing study limitations would enhance its impact:
- It comprehensively covers recent advances in virotherapy, summarizing findings from multiple studies and clinical trials. However, it lacks a discussion on potential biases or limitations in the studies reviewed.
- The paper effectively compares various oncolytic viruses and their combinations with other therapies. However, it could benefit from a more detailed comparison with non-virotherapy treatments for glioblastoma, to better highlight the advantages and limitations of virotherapy.
- I suggest to make the graph showing the mechanism of action of the viruses on oncolysis of the cells.
- Suggestions for Improvement: Include a section discussing potential limitations or biases in the studies reviewed. Expand on comparisons with alternative glioblastoma treatments to provide a more rounded view of the treatment landscape. A discussion on future directions or emerging research areas in oncolytic virotherapy would be beneficial.
- Grammar errors: 1) line 5: "ParvOryx01 were re-administered around the resection cavity." Is singular. 2) Line 21: "For 3 of 13 patients long-term survival of more than 46 months was observed." Improve clarity, 3) Line 23 "Cytotoxic T-lymphocytes (CTLs) recognize these peptides on APCs leading to the activa- tion of CTLs their migration to the tumor..." Punctuation, 5) Line “29 It worth noting that G207 con- verted immunologically “cold” tumors to “hot”..." Lacks completeness, 6) Line 29: "...immune surveillance is avoided due to incorrect work of immune checkpoints..." Different words should improve clarity, 7) Line 39 "...its DNA do not integrate into host genome..." DNA is singular.
Otherwise a nice paper, worth improvement and reconsidering after revisions.
Author Response
Thank you for your careful review of our manuscript. We tried to make changes in accordance with your suggestions. We added therapies and studies during the recruitment phase where possible. In addition, we added a separate paragraph where we highlighted the limitations, advantages and future directions for further research in the area under consideration. Grammar errors were have been corrected.
Changes in the text are indicated by color.